# Long-Read- and Short-Read-Based Whole-Genome Sequencing Reveals the Antibiotic Resistance Pattern of *Helicobacter pylori*

Limiao Hu,[a] Xi Zeng,[a] Qi Ai,[a] Caijuan Liu,[a] Xiaotuan Zhang,[b] Yajun Chen,[c] Logen Liu,[c,d] Guo-Qing Li[a,d,e]

[a]Department of Gastroenterology, the Second Affiliated Hospital, Hengyang Medical School, University of South China, Hengyang, China
[b]Department of Clinical Laboratory, The First Affiliated Hospital of Wenzhou Medical University, Wenzhou, China
[c]Clinical Research Center, the Second Affiliated Hospital, Hengyang Medical School, University of South China, Hengyang, China
[d]The Key Laboratory of Molecular Diagnosis and Precision Medicine in Hengyang, Hengyang, China
[e]The Clinical Research Center for Gastric Cancer in Hunan Province, Hengyang, China

Limiao Hu and Xi Zeng contributed equally to this work. Author order was determined both alphabetically and in order of increasing seniority.

**ABSTRACT** The rates of antibiotic resistance of *Helicobacter pylori* are increasing, and the patterns of resistance are region and population specific. Here, we elucidated the antibiotic resistance pattern of *H. pylori* in a single center in China and compared short-read- and long-read-based whole-genome sequencing for identifying the genotypes. Resistance rates of 38.5%, 61.5%, 27.9%, and 13.5% against clarithromycin, metronidazole, levofloxacin, and amoxicillin were determined, respectively, while no strain was resistant to tetracycline or furazolidone. Single nucleotide variations (SNVs) in the 23S rRNA and GyrA/B genes revealed by Illumina short-read sequencing showed good diagnostic abilities for clarithromycin and levofloxacin resistance, respectively. Nanopore long-read sequencing also showed a good efficiency in elucidating SNVs in the 23S rRNA gene and, thus, a good ability to detect clarithromycin resistance. The two technologies displayed good consistency in discovering SNVs and shared 76% of SNVs detected in the rRNA gene. Taking Sanger sequencing as the gold standard, Illumina short-read sequencing showed a slightly higher accuracy for discovering SNVs than Nanopore sequencing. There are two copies of the rRNA gene in the genome of *H. pylori*, and we found that the two copies were not the same in at least 26% of the strains tested, indicating their heterozygous status. Especially, three strains harboring a 2143G/A heterozygous status in the 23S rRNA gene, which is the most important site for clarithromycin resistance, were found. In conclusion, our results provide evidence for an empirical first-line treatment for *H. pylori* eradication in clinical settings. Moreover, we show that Nanopore sequencing is a potential tool for predicting clarithromycin resistance.

**IMPORTANCE** *Helicobacter pylori* resistance has been increasing in recent years. The resistance profile, which is important for empirical treatment, is region and population specific. We found high rates of resistance to metronidazole, clarithromycin, and levofloxacin in *H. pylori* in our center, while no resistance to tetracycline or furazolidone was found. These results provide a reference for local physicians prescribing antibiotics for *H. pylori* eradication. Nanopore sequencing recently appeared to be a promising technology for elucidating whole-genome sequences, which generates long sequencing reads and is time-efficient and portable. However, a relatively higher error rate of sequencing reads was also found. In this study, we compared Nanopore sequencing and Illumina sequencing for revealing single nucleotide variations in the 23S rRNA gene, which determines clarithromycin resistance, and we found that although there were a few false discoveries, Nanopore sequencing showed good consistency with Illumina sequencing, indicating that it is a potential tool for predicting clarithromycin resistance.

**KEYWORDS** Nanopore sequencing, *Helicobacter pylori*, antibiotic resistance, whole-genome sequencing, single nucleotide variation

Address correspondence to Guo-Qing Li, ligq1970@163.com, or Logen Liu, llg@fsyy.usc.edu.cn.

The authors declare no conflict of interest.

*H*elicobacter pylori colonizes the gastric mucosa of more than 50% of the world's population (1). *H. pylori* is a main risk factor for chronic active gastritis and peptic ulcer disease. The eradication of *H. pylori* enhances the treatment of ulcers, thereby making it a curable disease (2). The decline in the incidence of peptic ulcer disease in recent years is believed to be due to the successful eradication of *H. pylori* and a decrease in *H. pylori* infections (3). Furthermore, *H. pylori* is also a class I carcinogen as defined by the International Agency for Research on Cancer and is related to gastric cancer and mucosa-associated lymphoid tissue lymphoma (4). *H. pylori* eradication therapy, especially in high-risk populations (i.e., family history), reduced the incidence of gastric cancer and gastric cancer-related mortality (4, 5).

In clinical practice, only a few antibiotics such as clarithromycin (CLR), levofloxacin (LVX), metronidazole (MTZ), tetracycline (TCY), and amoxicillin (AMX) are proven to be effective in eradicating *H. pylori*. Generally, CLR-based empirical triple therapy has been administered to eradicate *H. pylori*. In areas where the rate of CLR resistance is >15%, several guidelines recommend bismuth-containing quadruple therapy (bismuth, a proton pump inhibitor, and two antibiotics) as the first-line therapy to eradicate *H. pylori* (6, 7). The eradication rate with the traditional triple therapy is approximately 75% (8), whereas bismuth-based quadruple therapy or the addition of bismuth to the traditional triple therapy greatly increased the eradication rates to >90% (9, 10). However, variations are observed worldwide. A large randomized controlled trial that recruited 184,786 participants and was performed in a region of China with a high incidence of gastric cancer showed that the overall eradication rate with the quadruple therapy (bismuth, omeprazole, TCY, and MTZ) is 72.9% (11). The failure to eradicate *H. pylori* is due mainly to increased resistance, although patient compliance and other factors such as drinking and smoking are also nonnegligible factors (12). Several studies reported the antimicrobial resistance (AMR) of *H. pylori* in recent years, and only a few studies reported a CLR resistance rate of <15% (13), a threshold suggested by most clinical guidelines. The resistance profiles are also different globally. Fernández-Reyes et al. (14) reported no *H. pylori* resistance to AMX and a high rate of resistance to CLR (24.2%) in northern Spain. Contrarily, Kouitcheu Mabeku et al. (15) reported a high rate of resistance (97.1%) to AMX and a low rate of resistance to CLR in Cameroon, Africa. Even in the same country, the resistance rates may also be different. Gao et al. (16) reported rates of resistance to CLR and LVX of 42.1% and 41.7%, respectively, in Beijing, China, while Wang et al. (17) reported rates of resistance of 31.0% and 56.0%, respectively, in northeastern China. Obtaining the local resistance pattern is important for physicians, especially for empirically choosing a suitable first-line prescription.

A2142G and A2143G mutations in the 23S rRNA gene are the most frequently reported mechanisms conferring resistance to clarithromycin. Mutations in the TCY-binding site of the 16S rRNA gene contribute to TCY resistance. Mutations in the penicillin-binding motif of penicillin-binding protein (PBP) confer AMX resistance, and resistance mediated by mobile genetic elements is rarely reported (18). Identifying the types of mutations using PCR or whole-genome sequencing (WGS) may guide the treatment of *H. pylori* infection. Previous studies focused mainly on second-generation sequencing (mainly the Illumina platform) to elucidate resistance mechanisms, and a few studies used third-generation sequencing technologies (long-read sequencing, i.e., Pacific Biosciences and Oxford Nanopore Technologies) to assemble the *H. pylori* genome (19–21). However, these technologies are used in limited strains, and the feasibility of long-read-based sequencing technology for elucidating the antibiotic resistance of *H. pylori* has not yet been evaluated. Here, we compared the two technologies for discovering single nucleotide variations (SNVs) and, thus, their diagnostic capabilities in antibiotic resistance.

## RESULTS

**Strains, demographics, and clinical features.** A total of 104 *H. pylori* strains were cultured from endoscopic gastric biopsy specimens collected from 300 patients in a tertiary hospital in central southern China. Quality checks suggested that there are contamination reads in the Illumina sequencing data for strain Hpfe091; thus, it was excluded from the SNV

**TABLE 1** Patient demographics and resistance to antibiotics of the corresponding strains

| Patient characteristic | No. of patients | No. (%) of patients with strain with resistance to: | | | |
|---|---|---|---|---|---|
| | | LVX | CLR | MTZ | AMX |
| Age (yrs) | | | | | |
| ~18–35 | 7 | 1 (14.3) | 2 (28.6) | 4 (57.1) | 0 |
| ~36–59 | 79 | 25 (31.6) | 33 (41.8) | 53 (67.1) | 12 (15.2) |
| ~60 | 18 | 3 (16.7) | 5 (27.8) | 7 (38.9) | 2 (11.1) |
| | | | | | |
| Sex | | | | | |
| Male | 69 | 14 (20.3) | 24 (34.8) | 44 (63.8) | 11 (15.9) |
| Female | 35 | 15 (42.9) | 16 (45.7) | 20 (57.1) | 3 (8.6) |
| | | | | | |
| Diagnosis | | | | | |
| Gastritis/duodenitis | 32 | 10 (31.3) | 14 (43.8) | 20 (62.5) | 7 (21.9) |
| Peptic ulcer | 65 | 16 (24.6) | 20 (30.8) | 40 (61.5) | 6 (9.2) |
| Gastric polyps | 6 | 3 (50) | 6 (100) | 4 (66.7) | 1 (16.7) |
| Gastric cancer | 1 | 0 | 0 | 0 | 0 |
| | | | | | |
| Total | 104 | 29 (27.9) | 40 (38.5) | 64 (61.5) | 14 (13.5) |

analysis. Data on the demographic characteristics of the 104 patients were collected, and there were 69 male and 35 female patients. Most of the patients were aged between 36 and 60 years and diagnosed as having a peptic ulcer or gastritis/duodenitis (Table 1).

**Antibiotic resistance phenotypes.** The patterns of antibiotic resistance of 104 cultured strains to five antibiotics (CLR, MTZ, LVX, TCY, and AMX) were determined (representative images are shown in Fig. S1 in the supplemental material) by an Etest. Twenty-three isolates (22.1%) were sensitive to the five antibiotics tested, and 81 isolates (77.9%) were resistant to at least one antibiotic. Forty isolates (38.5%) showed resistance to CLR (Fig. 1A), which was much higher than the threshold (15%) suggested by guidelines to empirically prescribe CLR-based triple therapy to eradicate *H. pylori*. One strain (1.0%) showed intermediate sensitivity, with an MIC of 0.5 μg/mL, and 63 strains (60.6%) were sensitive to CLR. In addition, the

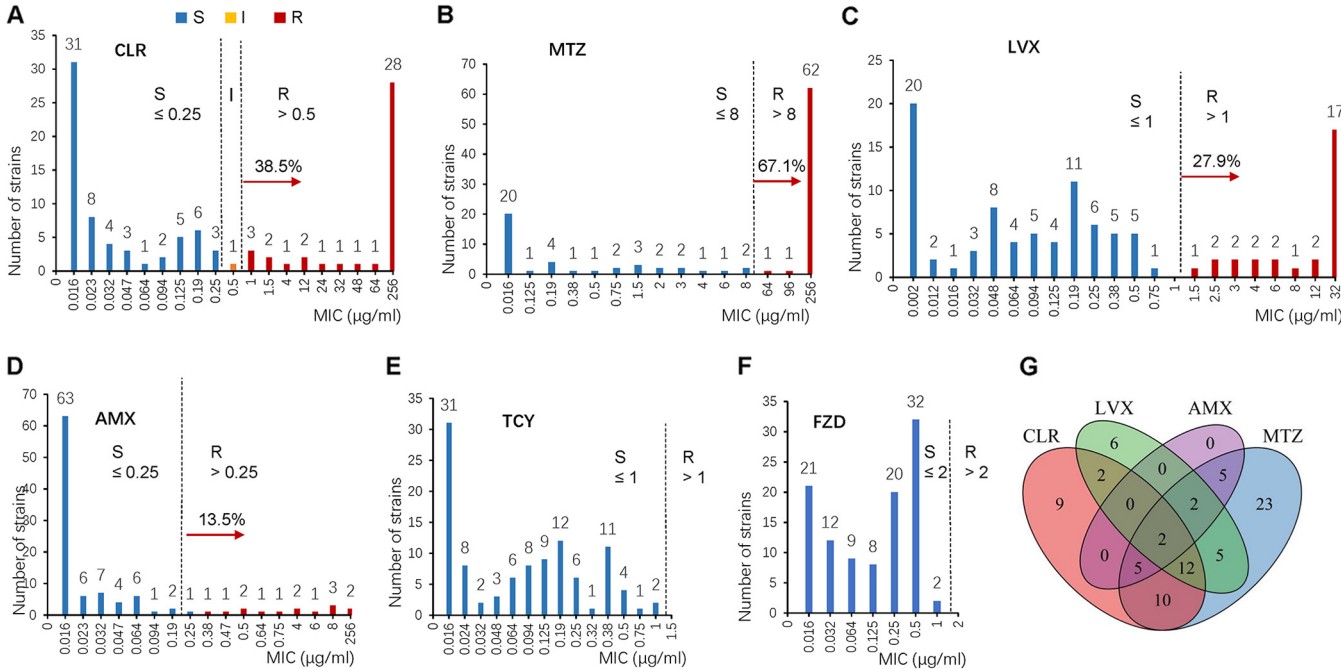

**FIG 1** Antibiogram of the 104 *H. pylori* strains. (A to E) Antibiotic susceptibility testing of *H. pylori* against CLR, MTZ, LVX, AMX, and TCY. The MIC values were determined using the Etest method. (F) Antibiotic susceptibility of *H. pylori* to FZD. The MIC values were determined using the agar dilution method. S, susceptible; I, intermediate; R, resistant. (G) Venn diagram showing the antibiotic resistance of *H. pylori*. Overlapping areas indicate multiple-drug resistance, and the numbers in the overlaps indicate the numbers of strains.

**TABLE 2** Diagnostic performance of selected mutations

| Mutation | No. of strains of phenotype | | Youden's index |
|---|---|---|---|
| | **Resistant** | **Sensitive** | |
| 23S rRNA (CLR) | | | 0.93 |
| Resistant | 37 | 0 | |
| Sensitive | 3 | 64 | |
| | | | |
| GyrA (LVX) | | | 0.67 |
| Resistant | 21 | 4 | |
| Sensitive | 8 | 71 | |

strains (64 strains; 61.5%) showed high-level resistance to MTZ (Fig. 1B). Twenty-nine strains (27.9%) were resistant to LVX (Fig. 1C), which suggests high-level resistance to this recently introduced antibiotic. Meanwhile, 14 strains (13.5%) were resistant to AMX (Fig. 1D), suggesting a relatively low rate of resistance to this antibiotic. No resistance to TCY was found for the 104 strains tested (Fig. 1E). Meanwhile, the pattern of resistance to furazolidone (FZD) was determined using agar dilution methods. Remarkably, no strains were resistant to FZD (Fig. 1F).

The resistance patterns among different diseases and genders seem to be different in this study. The rate of resistance to LVX is higher in female patients than in male patients ($P < 0.01$ by a chi-square test) (Table 1). Fifteen strains isolated from 35 female patients (42.9%) were resistant to LVX, while there were only 14 strains isolated from 69 male patients that were resistant to LVX (20.3%). Six strains isolated from gastric polyps were all resistant to CLR; 3 of these strains were resistant to LVX, and 4 of them were resistant to MTZ (Table 1).

CLR-based therapies are the first-line solutions to eradicate *H. pylori*, and multiple-antibiotic resistance, including clarithromycin resistance, may result in treatment failure. Therefore, we investigated the rates of multiple-antibiotic resistance of the strains. Of the 81 strains that showed resistance, 38 were resistant to a single antibiotic (CLR, LVX, or MTZ), whereas 43 strains were resistant to two or more antibiotics. Twenty-one strains exhibited resistance to three or more antibiotics (Fig. 1G), whereas two strains were resistant to four antibiotics (CLR, LVX, MTZ, and AMX). The detailed MIC values for each strain are listed in Table S1.

**Resistance genotypes revealed by Illumina sequencing.** To elucidate the mechanisms underlying resistance, we performed whole-genome sequencing of the 104 strains using the Illumina platform, and SNVs were analyzed by an assembly-based method. The results showed that 34 strains harbored the A2143G mutation; among them, 31 were resistant to CLR, whereas 3 strains (Hpfe063, Hpfe090, and Hpfe119) that were resistant to CLR harbored the A2142G mutation. Genotypic CLR susceptibility testing based on codon-positions 2142 and 2143 mutations of 23S rRNA showed a good diagnostic performance, with a Youden index value of 0.93 (Table 2).

Resistance to LVX and AMX is often conferred by mutations in the amino acid residues of proteins. To identify these mutations, tBLASTn analysis of the assembled genome was performed to search for GyrA and GyrB mutations. Of the 29 strains resistant to LVX, 16 strains harbored an N87K/I/D mutation in GyrA, most of which had MIC values of $>32$ μg/mL, much higher than the resistance cutoff value of 1 μg/mL (Table S1). Four strains harboring a D91G/I mutation showed low to intermediate MIC values, which were within the range of 1.5 to 6 μg/mL. Strain Hpfe114 harbored an A88V mutation and had an MIC value of 4 μg/mL. Meanwhile, four strains possessed an N87K/I/Y mutation and were sensitive to LVX. No other mutations in the quinolone resistance-determining region (QRDR) (A71 to Q110) were found. Furthermore, no resistant strains with GyrB mutations in the QRDR (E415 to S454) were observed. In addition, eight LVX-resistant strains could not be explained by the reported GyrA or GyrB mutations. Nevertheless, GyrA mutations showed a good diagnostic performance, with a Youden index value of 0.67 (Table 2).

A total of 14 strains were resistant to AMX. Mutations in PBP1A were reported to be the predominant cause of AMX resistance. A T593A/K/P/S mutation was found in five

AMX-resistant strains. However, 12 AMX-sensitive strains also possessed this mutation. Similarly, a G59S mutation was observed in 4 amoxicillin-resistant strains and 12 amoxicillin-sensitive strains. Eight resistant strains presented one or more mutations in the penicillin-binding motifs at the C-terminal region (22, 23), and five strains did not harbor mutations. These results suggested a poor phenotype-genotype relationship in the AMX resistance patterns and, possibly, a more complicated resistance mechanism in these strains.

We also checked the genotypes of resistance to MTZ. Consistent with previous reports (24–26), the mechanism of resistance to MTZ is more complicated, and no single mutation genotype could predict the resistance phenotype (data not shown).

**SNVs in the 23S rRNA gene detected using Nanopore sequencing.** SNVs in the 23S rRNA gene mediate resistance to CLR. Previous whole-genome sequencing studies focused on short reads to study the mutations. Furthermore, the feasibility of the use of Nanopore-based sequencing technologies, which are error prone but time efficient, for identifying SNVs in *H. pylori* has not yet been explored. In addition, consistency in the observed SNV genotypes and antibiotic resistance phenotypes was not evaluated. Here, we compared the performances of Nanopore sequencing and Illumina sequencing for revealing SNVs in the 23S rRNA gene and identifying mutation genotypes of antibiotic resistance phenotypes.

We first examined whether Nanopore sequencing can discover A2143G mutations found in the 34 *H. pylori* strains. Nanopore sequencing of the 104 strains was performed, and 98 strains generated reads with more than ~30× coverage in the 23S rRNA gene. Thus, the 98 strains were analyzed for A2143G and other mutations in the 23S rRNA gene (and its nearby 5S rRNA gene) using Clair3 software (referred to as the "Nanopore-Clair3 pipeline" below), which is a deep-learning-based variant caller, and pretrained with 30× data (27). A2143G mutations were detected in 34 strains with this Nanopore-Clair3 pipeline, 33 of which were also detected by the short-read-sequencing-based pipeline. The A2143G mutation of strain Hpfe126, which is resistant to CLR, was detected only by Nanopore sequencing. These results showed the good ability of Nanopore sequencing to identify SNVs that are related to antibiotic resistance.

We then compared the two sequencing technologies for discovering SNVs in other sites among these 98 strains. We found that the two technologies have a good correlation in the detection of SNVs. A total of 5,760 SNVs (including single nucleotide polymorphisms and small indels) were found in the 23S rRNA genes (and the nearby 5S rRNA genes) of the 98 strains using the two technologies, and 4,380 of them (76%) were discovered using both technologies. Meanwhile, 127 SNVs (2.2%) were detected only using the short-read-sequencing-based pipeline, whereas 1,253 (21.8%) were detected only using the Nanopore-Clair3 pipeline (Fig. 2A). The latter tended to find more SNVs, which may be due to the higher error rate of the Nanopore sequencing technology. Indeed, when the full-length 23S rRNA genes of 23 strains were sequenced with the Sanger method, of the 658 SNVs detected by Illumina- and Nanopore-based methods, 56 SNVs (8.5%) were falsely discovered by the Nanopore-based method, and only 1 SNV was falsely discovered by the Illumina-based method.

**Heterozygous status of rRNA.** Two copies of the 23S rRNA gene are present at different loci in the *H. pylori* genome. As shown in Fig. 2B, for *H. pylori* reference strain 26695, their surrounding genomic contents are different. We sought to find out if the two 23S rRNA "alleles" are different, causing a heterozygous status, which may cause resistance with low to moderate MIC values and may be intermediate products of resistance evolution (28). Due to the high identity of these two copies, short-read assemblies usually collapse them into a single locus, which makes it almost impossible to identify the heterozygosity status of the 23S rRNA gene. To compare the sequences of the two 23S rRNA loci and explore the heterozygous status, we performed a mapping-based SNV-calling algorithm. The Illumina reads were aligned to a reference rRNA gene, sorted, and piled up using SAMtools software, and the SNVs of each site were then identified using VarScan (referred to as the "Illumina-VarScan pipeline" below). SNVs with a >5% (and a <95%) frequency, 100× coverage, and at least 10 reads covering each "allele" were screened out as true heterozygous sites. The results showed that 26 strains possessed a heterozygous status of SNVs in the 23S rRNA gene (Table 3 and Table S2). Importantly, two strains (Hpfe061 and Hpfe092)

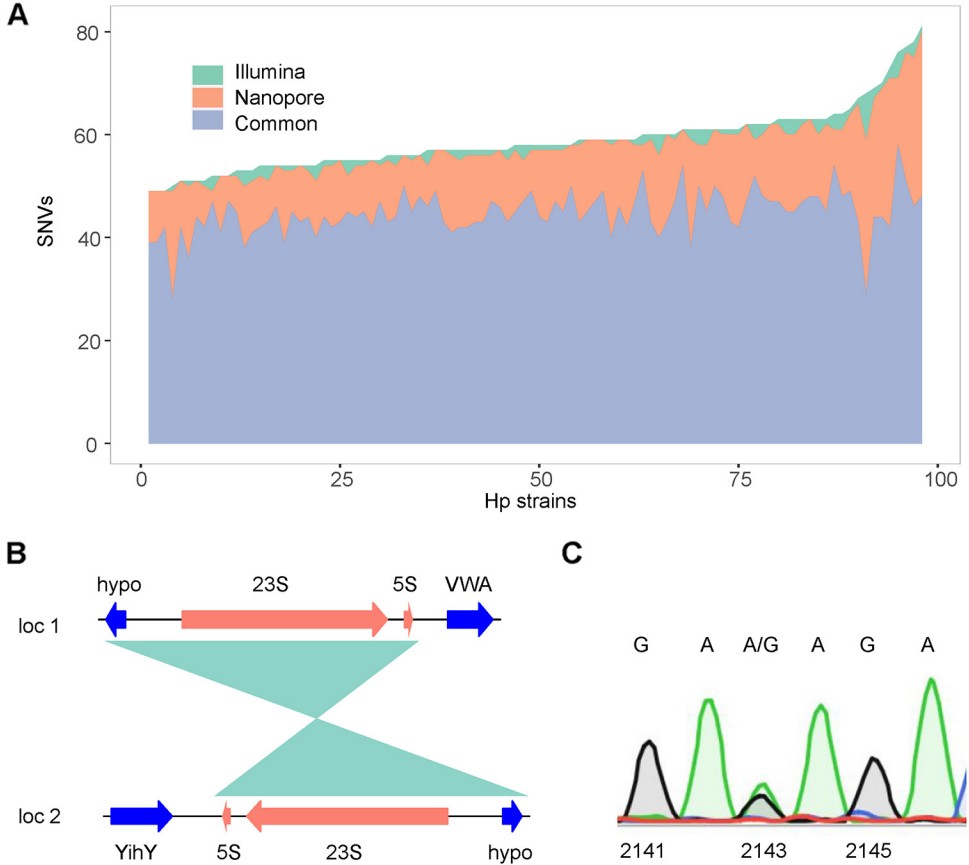

**FIG 2** Consistency of Illumina- and Nanopore-based sequencing in discovering SNVs of the *H. pylori* 23S rRNA gene and its heterozygous status. (A) Stacked area graph showing the SNVs identified using Illumina and Nanopore technologies. SNVs of 98 strains were determined using the Illumina-VarScan pipeline or the Nanopore-Clair3 pipeline. The SNVs determined using the two pipelines were then compared. Common, SNVs detected using both pipelines; Nanopore, SNVs detected using only the Nanopore-Clair3 pipeline; Illumina, SNVs detected using only the Illumina-VarScan pipeline. (B) The genomic contents of the two copies of the 23S rRNA gene are different. In locus 1, the gene downstream of the 23S-5S gene is the gene encoding a von Willebrand/Integrin A Domain containing protein (VWA), while in locus 2, the 23S-5S gene is located in the complementary strand, and the downstream gene is the gene encoding YihY. The shaded region is the homologous region. hypo: hypothetical protein. (C) SNVs of 23S rRNA genes confirmed by colony PCR and Sanger sequencing. Individual colonies of *H. pylori* grown on plates were picked as the templates, and PCR of the 23S rRNA gene was performed. The amplified fragments were subjected to Sanger sequencing. There are both A and G fluorescence signals (green and black, respectively) at position 2143 of the 23S rRNA gene of strain Hpfe061.

showed a heterozygous status of A and G at nucleotide 2143 of the two 23S rRNA copies, both of which are resistant to CLR, suggesting a transition resistance phenotype. To further confirm that the heterozygous status detected is the result of two different alleles of a single clone and not a manifestation of mixed multiple colonies, colony PCRs of the 23S rRNA gene followed by Sanger sequencing were performed on the strains. As shown in Fig. 2C, both A and G signals were detected at position 2143 of the 23S rRNA gene of strain Hpfe061, indicating a true heterozygous status. Besides A2143R, other heterozygous sites were also verified by Sanger sequencing.

We next checked the consistency of the two pipelines in discovering SNVs in the 16S rRNA gene. 16S rRNA mutations are responsible for tetracycline resistance. In this survey, the A926C mutation, at a site at which tetracycline binds, was found in a single strain, Hpfe080, and the MIC value of this strain was 1.5 $\mu$g/mL, an intermediate state for antibiotic susceptibility testing. However, multiple mutations at other sites were found by the two pipelines (Fig. 3A and Table S2). A total of 864 (71%) SNVs were found by both pipelines, 327 SNVs (27%) were detected only using Nanopore-Clair3, and 23 (2%) were detected only by the Illumina-VarScan pipeline (Fig. 3A).

Similarly, the two copies of the 16S rRNA gene are displaced in two different loci in

**TABLE 3** Heterozygous status of the 23S rRNA gene and CLR resistance[a]

| Strain | Heterozygous site | VarFreq (%) | No. of reads | CLR MIC ($\mu$g/mL) | Phenotype |
|---|---|---|---|---|---|
| Hpfe0002 | C1723Y | 47.33 | 572/514 | >256 | Res |
| Hpfe0020 | G15R | 51.34 | 344/363 | 0.19 | Sen |
| Hpfe0032 | C2811Y | 50.03 | 836/838 | 0.125 | Sen |
| Hpfe058 | A720R | 47.19 | 1,823/1,629 | 0.064 | Sen |
| | C2916Y | 44.61 | 1,620/1,307 | | |
| | T2921Y | 54.55 | 1,325/1,590 | | |
| Hpfe061 | **A2143R** | 47.67 | 877/799 | >256 | Res |
| Hpfe064 | C14Y | 49.56 | 345/339 | >256 | Res |
| Hpfe089 | C2195Y | 49.51 | 618/608 | 1.5 | Res |
| Hpfe092 | **A2143R** | 48.61 | 1,222/1,156 | 64 | Res |
| Hpfe106 | G1056R | 48.86 | 988/945 | 256 | Res |
| Hpfe124 | G2599R | 51.52 | 1,195/1,270 | <0.016 | Sen |
| Hpfe125 | G156R | 47.94 | 782/720 | <0.016 | Sen |

[a]Selected strains are shown. For full information, see Table S2 in the supplemental material. R is A or G, and Y is C or T. The number of reads is the number of reads that support the wild-type nucleotide or the mutated nucleotide at that site. VarFreq, frequency of the variant; Res, resistant to CLR; Sen, sensitive to CLR. A2143R heterozygosity, which may mediates CLR resistance is highlighted with boldface.

the chromosome, and their surrounding genes are different as well (Fig. 3B). A heterozygous status of the two 16S rRNA copies was found for 2 strains (Hpfe113 and Hpfe126) via the Illumina-VarScan pipeline. Two sites with a heterozygous status, T92K (where K is T or G) and C189Y, were found in strain Hpfe126. The heterozygous status was also confirmed by Sanger sequencing. As exemplified in Fig. 3C, C189Y was detected by Sanger sequencing.

## DISCUSSION

*H. pylori* is a class I carcinogen, and eradication is necessary once infection is confirmed (29). *H. pylori* is a microaerophilic bacterium that has strict culture conditions, and many clinical laboratories are unable to perform this test. Furthermore, the culture success rate is relatively low; in this study, we observed a positivity rate of 36.1%, while previous studies reported positivity rates ranging from 26.5% to 55.4% (19, 20, 24, 30). Therefore, clinical guidelines have suggested empirical treatment based on local epidemiology and antibiotic resistance data unless several trials fail (6, 29). Local epidemiology and resistance profiles are important in eradicating *H. pylori*. Interestingly, the AMR pattern in China differs from that in Europe or Africa. In Europe and Japan, the reported rates of resistance to CLR and MTZ were relatively low, possibly due to the effective antibiotic control policies applied (13, 31). In China, AMX is frequently used as self-medication by patients with upper respiratory infections. However, in this study, we observed a relatively low rate of resistance to AMX (13.5%), which is also consistent with the results of other reports from China: a similar resistance rate of 18.52% was reported in southern China (20), and AMX resistance was not observed in children from southwestern China (30). Some African countries showed 97.1% resistance to AMX (22, 30). Meanwhile, the rate of resistance to MTZ is very high worldwide (13). In general, the antibiotic resistance of *H. pylori* is increasing compared to that 10 or 20 years ago (20, 31).

The resistance genotypes of *H. pylori* differ globally, as point mutations are believed to emerge *de novo* in *H. pylori*, as opposed to other bacteria that obtain resistance to antibiotics by the horizontal transfer of mobile genetic elements. In the present study, A2142G and A2143G were the common mutations found in CLR-resistant *H. pylori* strains. Four CLR-resistant strains, including one strain identified using Nanopore sequencing, harbored the A2142G mutation. This observation is consistent with reports from China and Asian countries such as Malaysia and Iran (32, 33). In other studies, A2142G mutations were more common, and the prevalence of the A2142G/C mutation was as high as 30% in Zurich, Switzerland (25). In addition, other mutations in domain V of the 23S rRNA gene were also reported to confer resistance in a few strains (34). However, no significant relationship of

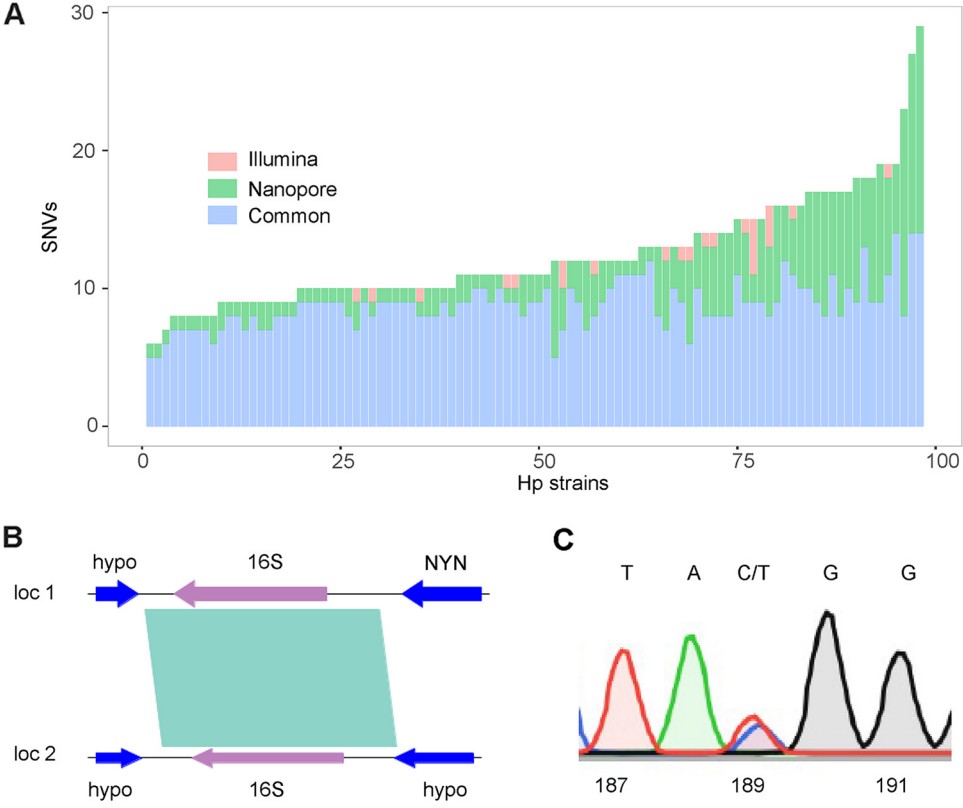

**FIG 3** Consistency of Illumina- and Nanopore-based sequencing in discovering SNVs of the *H. pylori* 16S rRNA gene and its heterozygous status. (A) Stacked bar graph showing the SNVs identified in the 16S rRNA gene using Illumina and Nanopore technologies. Nanopore, SNVs detected using only the Nanopore-Clair3 pipeline; Illumina, SNVs detected using only the Illumina-VarScan pipeline; Common, SNVs detected using both pipelines. (B) The genomic contents of the two copies of the 16S rRNA gene are different. In locus 1, the genes surrounding the 16S rRNA gene are genes encoding a hypothetical protein and an Nedd4-BP1/YacP-like Nuclease domain containing protein (NYN), while in locus 2, there are two other genes encoding hypothetical proteins (hypo). The shaded region is the homologous region. (C) SNVs of the 16S rRNA gene confirmed by colony PCR and Sanger sequencing. There are both C and T fluorescence signals (blue and red, respectively) at position 189 of the 16S rRNA gene of strain Hpfe126, consistent with the results of both the Illumina-VarScan and Nanopore-Clair3 pipelines.

these sites to CLR resistance was observed in our study, possibly due to the low frequency of these mutations in the population.

Nanopore read-based SNV calling also identified mutations at positions 2142 and 2143 that are responsible for CLR resistance, indicating that it is a good tool for the genotyping of CLR resistance. However, generally speaking, it resulted in more false-positive results than Illumina read-based SNV calling, which may be due to the higher error rate of this technology. Still, there are also many other factors that influence the accuracy of the deep-learning-based Nanopore-Clair3 SNV-calling pipeline, including the sequencing machine (MinION, PromethION, or GridION), the Guppy base caller version and mode used, and the training data (27). With continuous development, the sequencing accuracy of Nanopore sequencing will progressively improve, and the latest chemistry claimed an accuracy of up to 98.3% (https://nanoporetech.com/accuracy). The improved accuracy may further make it feasible for SNV detection for *H. pylori* antibiotic resistance prediction.

Differences in the sequences of the two copies of the 23S rRNA gene in *H. pylori* have been rarely investigated. Marques et al. (34) reported the possible hybrid status of A2142R (where R is A or G) in two strains of *H. pylori*, but the G mutation in one of the strains was observed in only 10 sequencing reads. Short-read- and assembly-based SNV-calling methods usually collapse the 23S rRNA gene into a single locus; thus, it is challenging to identify its heterozygous status. In this study, SPAdes software assembled 91 full-length 23S rRNA genes from the 108 strains, and no strain with two 23S rRNA genes was found in the

assemblies (data not shown). To determine the hybrid status of the 23S rRNA gene of *H. pylori*, we applied mapping-based SNV calling using short reads, and we found that an SNV hybrid status of the 23S rRNA gene existed in 25 strains, including 2 strains harboring the A2143R mutation (Table 3; see also Table S2 in the supplemental material). Moreover, the frequency was >5% of the reads and coverage. These results were further confirmed using PCR and Sanger sequencing of the 23S rRNA gene.

In summary, we describe the resistance pattern of *H. pylori* in central southern China with data from a single center and elucidate the molecular resistance pattern using combined second-generation Illumina short-read sequencing and Nanopore long-read sequencing. We completed the whole genomes of selected strains and evaluated the hybrid state of 23S rRNA mutations. Our study indicates that the rate of CLR resistance is much higher than the threshold of 15% in central southern China. We also found a high rate of resistance to MTZ, which should be avoided when selecting empirical therapy. Relatively low rates of resistance to AMX and LVX were found. Based on 98 strains subjected to Nanopore sequencing and SNV calling, the technology has showed to be an effective tool for predicting CLR resistance as good as short-read-based sequencing.

## MATERIALS AND METHODS

**H. pylori isolates.** Patients with a positive [$^{13}$C]urea breath test (UBT) were recruited for this research. Gastric specimens obtained using endoscopy were homogenized and spread onto Columbia blood agar plates (7% defibrinated sheep blood) supplemented with 2.5 $\mu$g/mL amphotericin B, 16 $\mu$g/mL cefsulodin, 20 $\mu$g/mL trimethoprim, and 6 $\mu$g/mL vancomycin (35). Subsequently, the plates were placed in a tri-gas incubator with 5% $CO_2$ and 6% $O_2$. After 3 to 5 days, the plates were observed for *H. pylori* growth, and Gram staining was then performed to confirm *H. pylori* morphology. A single clone was then chosen and spread onto another fresh plate for further analysis.

**Antibiotic susceptibility test.** *H. pylori* cells cultured on Columbia blood agar plates were carefully scraped with cotton swabs and adjusted to a McFarland standard of 3 using Columbia broth. Next, the bacterial suspension was spread onto new Columbia blood agar plates with Etest strips for CLR, LVX, AMX, MTZ, and TCY (Liofilchem, Italy). As there are no commercial Etest strips available, antibiotic susceptibility testing against furazolidone (FZD) was performed using an agar dilution method. Briefly, 5 $\mu$L of an *H. pylori* suspension (McFarland standard of 3) was inoculated onto Mueller-Hinton blood agar plates containing different concentrations of FZD (ranging from 0.016 to 8 $\mu$g/mL). The growth of *H. pylori* was observed after 3 days of incubation in a tri-gas incubator. *H. pylori* standard strains 26695 and ATCC 43504 were used as the controls. The interpretation of resistance was performed according to European Committee on Antimicrobial Susceptibility Testing (EUCAST) guidelines and recent reports (30).

**Genomic DNA extraction, library construction, and sequencing.** The genomic DNA of *H. pylori* was extracted using a bacterial genomic DNA kit (Tiangen, China). The quality and quantity of the extracted DNA were examined using agarose gel electrophoresis and a Qubit double-stranded DNA (dsDNA) high-sensitivity (HS) assay kit (Thermo Fisher, USA), respectively. The Illumina sequencing library was prepared using the NEBNext Ultra DNA library prep kit (New England BioLabs [NEB], USA), according to the manufacturer's instructions. Nanopore long-read sequencing was performed as described previously (36). The library was constructed using the SQK-RBK004 rapid barcoding kit (Oxford Nanopore Technologies, UK). Briefly, 400 ng of quality- and quantity-checked genomic DNA was incubated with the fragmentation mix at 30°C for 1 min for cleavage and barcoding, purified using AMPure XP beads (Beckman Coulter, USA), incubated with rapid adapter (RAP) reagent to add the sequencing adapter, mixed with sequencing buffer, and loaded onto a MinION sequencer (Oxford Nanopore Technologies, UK). Sequencing data were obtained in the subsequent 24 h.

**Assembly-based SNV analysis.** Raw Illumina reads were quality checked and filtered using Trimmomatic v0.39 (37), and the resulting clean reads were *de novo* assembled using SPAdes v0.14 (38). To confirm that the sequenced species are *H. pylori*, the average nucleotide identity (ANI) among the 104 strains and reference strain 26695 was calculated using FastANI, which showed that the identities are all higher than 95% (see Fig. S2 in the supplemental material). The assemblies were then subjected to BLASTn analysis against the 23S rRNA gene sequence of UA802 (GenBank accession number U27270). SNVs were generated by analyzing the output file. For single-amino-acid mutations that are responsible for antibiotic resistance, tBLASTn analysis was performed between the assemblies and the reference protein sequences (GyrA, GyrB, and Rpl22) of *H. pylori* strain 26695 (GenBank accession number NC_000915). Mutations were extracted from the BLAST output file. Mutations in other genes were identified similarly.

**Mapping-based SNV calling.** The clean Illumina reads were aligned to the 23S or 16S rRNA gene using bowtie2 software (39). Sorting, indexing, and pileup were performed using SAMtools (40). Variant calling was performed using VarScan (28). For SNV calling from Nanopore long reads, the reads were base called using Guppy (https://nanoporetech.com/community), and quality filtering was performed with the parameter setting "-Q 0.7" during base calling, which achieved an ~85% accuracy of the base-called reads. The filtered reads were mapped to the 23S or 16S rRNA gene using minimap2 (41) and sorted using SAMtools. The generated bam format files were subjected to analysis using Clair3 (27), which integrated the pileup and the full alignment from SNV calling and phasing.

**Statistical analysis.** A chi-square test (or Fisher's exact test, as appropriate) was performed to compare the differences in antibiotic resistance among different patient demographic features. Youden's index values were calculated when evaluating the diagnostic ability of selected mutations detected by WGS for predicting antibiotic resistance. All analyses were performed in the R programming language. A *P* value of <0.05 was considered statistically significant.

**Ethics approval.** This study was approved by the Medical Ethics Committee of the Second Affiliated Hospital, University of South China. Written consent was obtained from all subjects.

**Data availability.** The complete genomic sequences hybrid assembled using Unicycler v0.46 and raw Illumina sequencing data are stored in the NCBI database under BioProject accession number PRJNA816422. The raw fast5 files for Nanopore sequencing were demultiplexed with a Python script and are stored in the SRA database under the same BioProject accession number. The scripts used are available from GitHub (https://github.com/liu2005678/Hpresis).

## SUPPLEMENTAL MATERIAL

Supplemental material is available online only.
**SUPPLEMENTAL FILE 1**, XLSX file, 0.02 MB.
**SUPPLEMENTAL FILE 2**, PDF file, 0.7 MB.
**SUPPLEMENTAL FILE 3**, XLSX file, 0.01 MB.

## ACKNOWLEDGMENTS

This study was supported by grants from the Science and Technology Department of Hunan Province (grant numbers 2021SK4029 and 2021ZK4259), the Natural Science Foundation of Hunan Province (grant numbers 2021JJ40483, 2019JJ50534, and 2022JJ30517), the Education Department of Hunan Province (grant number 19B499), and the Science and Technology Bureau of Hengyang (grant numbers 2021QZY020 and 202010041574).

We have no conflicts of interest.

G.-Q.L. and L.L. conceived and designed the research. L.H., X.Z., Q.A., and C.L. collected the species and performed the experiments. L.H., X.Z., Y.C., and L.L. analyzed the data. L.H., X.Z., and Y.C. drafted the paper. L.L. and G.-Q.L. revised the manuscript, and all authors approved the final version of the manuscript.

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
