## [Reviewer comments · Microbiology Spectrum]

Microbiology Spectrum

Long Reads and Short Reads Based Whole Genome Sequencing Revealed the Antibiotic Resistance Pattern of *Helicobacter pylori*

Limiao Hu, Xi Zeng, Qi Ai, Caijuan Liu, Xiaotuan Zhang, Yajun Chen, Logen Liu, and Guoqing Li

Corresponding Author(s): Guoqing Li, University of South China

Review Timeline:

Submission Date:	November 6, 2022
Editorial Decision:	February 16, 2023
Revision Received:	March 28, 2023
Accepted:	March 29, 2023

Editor: Aude Ferran

Reviewer(s): Disclosure of reviewer identity is with reference to reviewer comments included in decision letter(s). The following individuals involved in review of your submission have agreed to reveal their identity: Teresa Alarcón (Reviewer #1); Xiaohua Long (Reviewer #2)

Transaction Report:

DOI: <https://doi.org/10.1128/spectrum.04522-22>

February 16, 2023

Prof. Guoqing Li
The second affiliated Hospital, University of South China
Ave. Jiefang 35
Hengyang 421001
China

Re: Spectrum04522-22 (Long Reads and Short Reads Based Whole Genome Sequencing Revealed the Antibiotic Resistance Pattern of *Helicobacter pylori*)

Dear Prof. Guoqing Li:

Thank you for submitting your manuscript to Microbiology Spectrum. The two reviewers mentioned that nanopore produced quite a lot of errors. You should make this clear in the text. You must also answer all questions asked by the reviewers.

Link Not Available

Sincerely,

Aude Ferran

Journals Department
Reviewer comments:

Reviewer #1 (Comments for the Author):

In this study two whole genome sequencing technologies were compared to study mutations related to antimicrobial resistance in *H. pylori*.

Some comments:

- Please check the information about the number of strains sequenced as they seem to be 104 according to the whole manuscript but according to Lines 114 and 115 "Four Hp strains exhibited very slow growth and thus, AST was not performed in these strains; however, whole genome sequencing was still performed."
- Table 1 should include an extra row with the total number of strains tested and resistance percentages without subgroups.
- Please include the legends of supplementary tables and figures
- Please include a section in methods describing the statistical test used
- In the supplementary table some data has a ?. Please clarify the meaning (for example Hpfe076, female, 43, > 32 ?)
- The strains Hpfe024 has a levofloxacin MIC very low, < 0.002 S, but it has a mutation by WGS. Was the E test repeated?
- Mutation in PBP1A should be included in supplementary table both in AMX susceptible or resistance and the genotypes obtained for metronidazole resistance should also be included. Although the results may not seem relevant they are interesting for authors studying the same subject.

Reviewer #2 (Comments for the Author):

The relationship between 23SrRNA mutation(A2143G,A2142G) and clarithromycin phenotype resistance has been well clarified by many researchers, however in this paper the A2143G mutation was still found in a CLR-S strain, the author did not mention (detected by Nanopore or Illumina sequencing) or discuss this discovery. Is the mutation A2143G in a CLR-S strain(MIC 0.25), a false among the 8.5%?

In the discussion part, this paper were not writtend logically nor more powerfully (eg,discussed on amoxicillin rather than clarithromycin).

Errors, Eg, Line 262 should be writtend as 13C-urea breath test(UBT).

Line79-82, the overall eradication rate of PPI+Bismuth+metronidazole+tetracyclin regime is only 72.3% (ITT)[11], which is too low to be an acceptable regime as David G.Y. suggested , in fact this low eradication rate is due to patient's low compliance (such as missed doses, drink or smoke)rather than antibiotic resistance in the referred paper, so the author should state more rigorously to improve the evidence.

Antibiotic resistance differs in different background, the author had better display whether the strains obtained from patient accepting initial or rescued therapy.

Staff Comments:

Preparing Revision Guidelines

Please return the manuscript within 60 days; if you cannot complete the modification within this time period, please contact me. If you do not wish to modify the manuscript and prefer to submit it to another journal, please notify me of your decision immediately so that the manuscript may be formally withdrawn from consideration by Microbiology Spectrum.

Reviewer comments:

Reviewer #1 (Comments for the Author):

In this study two whole genome sequencing technologies were compared to study mutations related to antimicrobial resistance in *H. pylori*.

Some comments:

- Please check the information about the number of strains sequenced as they seem to be 104 according to the whole manuscript but according to Lines 114 and 115 "Four Hp strains exhibited very slow growth and thus, AST was not performed in these strains; however, whole genome sequencing was still performed."

Answer: Thanks for carefully reviewing the manuscript and pointing out the issue. Because the four strains were not included in the following genotype and phenotype analysis and no information of the 4 stains were used, we decide to delete the sentence to avoid confusion, and change "108" to "104" in the revised version of manuscript (Line 112).

- Table 1 should include an extra row with the total number of strains tested and resistance percentages without subgroups.

Answer: An extra row was added in Table 1.

- Please include the legends of supplementary tables and figures

Answer: Legends for the Supplementary Figures and Tables have been added accordingly. (Supplementary PDF file).

- Please include a section in methods describing the statistical test used

Answer: Thanks for suggestion. A section of statistical analysis has been added in the revised manuscript (Lines 323-329).

- In the supplementary table some data has a ?. Please clarify the meaning (for example Hpfe076, female, 43, > 32 ?)

Answer: We are sorry for the carelessness. We have revised the table accordingly.

- The strains Hpfe024 has a levofloxacin MIC very low, <0.002 S, but it has a

mutation by WGS. Was the E test repeated?

Answer: Thanks for the reminder. We routinely repeat two times for the E-test. At the reminder of the reviewer, we repeated the E-test again and performed sanger sequencing of the strain which verified the N87K mutation and sensitivity to levofloxacin. The QRDR region mutation of *gryA* was not always consistent with antibiotic resistant phenotype, as indicated in this research that the diagnostic Youden's index of 0.671, and consistent with previous reports (for example, PMID: 33374988, in which three strains with MIC values of 0.008, 0.023, 0.004 were reported with N87I, N87K and N87T mutations respectively. and other reports, PMIDs: 27454429, 32545318, 35124867). It worth noting and interesting that the MIC < 0.002 here is even lower than the reports, which we will keep in mind.

- Mutation in PBP1A should be included in supplementary table both in AMX susceptible or resistance and the genotypes obtained for metronidazole resistance should also be included. Although the results may not seem relevant they are interesting for authors studying the same subject.

Answer: We have added information of the genotype related to AMX and MTZ to supplementary Table 1.

Reviewer #2 (Comments for the Author):

The relationship between 23S rRNA mutation (A2143G, A2142G) and clarithromycin phenotype resistance has been well clarified by many researchers, however in this paper the A2143G mutation was still found in a CLR-S strain, the author did not mention (detected by Nanopore or Illumina sequencing) or discuss this discovery. Is the mutation A2143G in a CLR-S strain (MIC 0.25), a false among the 8.5%?

Answer: Thanks for carefully reviewing the manuscript and providing the thoughtful suggestion. In the original manuscript, we described that possible contamination reads might exist for Hpfe091, which was sensitive to CLR (MIC 0.25), but detected with A2143G mutation by the Illumina sequencing data. We have re-sequenced the strain with Illumina platform and found no A2142G or A2143G mutation. Accordingly, we have updated the data in Tables 2 of the revised manuscript.

In the discussion part, this paper were not written logically nor more powerfully (eg, discussed on amoxicillin rather than clarithromycin).

Answer: Thanks for your helpful comment on the Discussion section. Accordingly, we have modified the section to pay more attention to CLR in the section of the revised manuscript (paragraph two of Discussion section. Lines 351-362).

Firstly, we discussed the epidemiology of Hp resistance that we found in the present study, with reference to previous data. Secondly, we discussed the relationship of A2142G, A2143G mutations of 23S rRNA to CLR resistance. Thirdly, we discussed the error rate of Nanopore sequencing and its potential in elucidating point mutations mediated antibiotic resistance. Fourthly, we discussed the heterozygous status and its possible impact of 23S rRNA.

We hope that the revised Discussion section is now acceptable.

Errors, Eg, Line 262 should be written as 13C-urea breath test (UBT).

Answer: We have changed accordingly.

Line 79-82, the overall eradication rate of PPI+Bismuth+metronidazole+tetracycline regime is only 72.3% (ITT)[11], which is too low to be an acceptable regime as David G.Y. suggested, in fact this low eradication rate is due to patient's low compliance (such as missed doses, drink or smoke) rather than antibiotic resistance in the referred paper, so the author should state more rigorously to improve the evidence.

Answer: Thanks for your knowledgeable comment on the low eradication rate obtained in the randomized controlled trial in China. We have made some

modifications in the revised manuscript accordingly (Lines 82-84). The revised portion has been highlighted, which echoes your opinion on the multi-factorial contribution to low OER of PPI+Bismuth+metronidazole+tetracyclin regimen.

Antibiotic resistance differs in different background, the author had better display whether the strains obtained from patient accepting initial or rescued therapy.

Answer: We do appreciate your insightful suggestion on “better display whether the strains obtained from patient accepting initial or rescued therapy”. However, it is very difficult for us to obtain the information retrospectively. Nevertheless, we will keep your suggestion in mind and incorporate into our future study relevant to this topic.

March 29, 2023

Prof. Guoqing Li
University of South China
Ave. Jiefang 35
Hengyang 421001
China

Re: Spectrum04522-22R1 (Long Reads and Short Reads Based Whole Genome Sequencing Revealed the Antibiotic Resistance Pattern of *Helicobacter pylori*)

Dear Prof. Guoqing Li:

Your manuscript has been accepted, and I am forwarding it to the ASM Journals Department for publication. You will be notified when your proofs are ready to be viewed.

Sincerely,

Aude Ferran
Editor, Microbiology Spectrum
